# Equity and Opportunities in Lung Cancer Care—Addressing Disparities, Challenges, and Pathways Forward

**DOI:** 10.3390/cancers17081347

**Published:** 2025-04-17

**Authors:** Dena G. Shehata, Jennifer Megan Pan, Zhuxuan Pan, Janani Vigneswaran, Nicolas Contreras, Emily Rodriguez, Sara Sakowitz, Jessica Magarinos, Sara Pereira, Fatima G. Wilder, Ammara A. Watkins

**Affiliations:** 1Division of Thoracic and Cardiovascular Surgery, Lahey Hospital and Medical Center, Burlington, MA 01805, USA; dena.shehata@lahey.org; 2Department of General Surgery, Beth Israel Deaconess Medical Center, Boston, MA 02115, USA; jmpan@bidmc.harvard.edu; 3UMass Chan Medical School, Worcester, MA 01655, USA; zhuxuan.pan@umassmed.edu; 4Department of Thoracic Surgery, University of Utah, Salt Lake City, UT 84132, USA; janani.vigneswaran@gmail.com (J.V.); u6035392@umail.utah.edu (N.C.); jessicamagarinos@gmail.com (J.M.); sara.pereira@hsc.utah.edu (S.P.); 5School of Medicine, The Johns Hopkins University, Baltimore, MD 21218, USA; erodri47@jhmi.edu; 6David Geffen School of Medicine, University of California, Los Angeles, CA 90024, USA; ssakowitz@mednet.ucla.edu; 7Division of Thoracic Surgery, Department of Surgery, Brigham and Women’s Hospital, Boston, MA 02115, USA; fwilder@bwh.harvard.edu

**Keywords:** lung cancer disparities, social determinants of health, lung cancer screening, race, ethnicity, survival, surgical treatment

## Abstract

Lung cancer is the leading cause of cancer-related mortality in the United States, with racial and ethnic minorities experiencing worse outcomes due to disparities in screening, diagnosis, and treatment. This review aims to identify the underlying causes of these disparities and explore strategies to improve lung cancer care for underserved populations. The findings highlight gaps in screening and treatment, suggest policy changes, and discuss targeted interventions to improve access to care and reduce health disparities in lung cancer outcomes.

## 1. Introduction

Lung cancer is the leading cause of cancer-related deaths in the United States, with a disproportionate impact on racial and ethnic minorities [1]. After a lung cancer diagnosis, the estimated 5-year survival rate is 26.7% [2,3] across all stages. The reasons for poor survival are multifaceted and are related to tobacco use, late-stage diagnosis, and underutilization of effective treatments [4]. County-level spatial analyses have highlighted that social determinants, including income inequality, education level, and rurality, are tightly linked to geographic disparities in cancer mortality [5].

It is evident in the literature that race and gender impact lung cancer diagnosis and treatment. The incidence of lung cancer is 40.7 per 100,000 in White individuals and 43.5 per 100,000 in Black individuals [3,4]. Studies have found that Black patients develop lung cancer earlier than White patients, with a median age of 67 compared to 70 [4]. Of the patients with lung cancer outside of the screening age range (50–80 years old), Black patients were almost twice as likely as White patients to be diagnosed (14.2 vs. 8.2%, *p* < 0.05) [1,6]. Furthermore, the rates of lung cancer death are disproportionate. Death rates are estimated at 35.1% for non-Hispanic Whites, 33.4% for non-Hispanic Blacks, 18.5% for Asian/Pacific Islanders, 26.3% for American Indian/Alaska Natives, and 14.2% for Latinos. Gender also has an impact on mortality rates. From 2016 to 2020, mortality rates were highest among Black males (51.0 per 100,000), followed by White males (44.7 per 100,000; 14% lower) and American Indian/Alaska Native males (39.9 per 100,000; 28% lower) [7]. Among women, White women had the highest death rates (32.8 per 100,000), followed by American Indian/Alaska Native women (29.9 per 100,000; 10% lower) and Black women (27.8 per 100,000; 18% lower) [7].

Elements of social determinants of health (SDOH) play an important role in these disparities. Socioeconomic status (SES) significantly impacts lung cancer disparities and influences incidence, treatment, and outcomes (Figure 1). Lower SES is associated with higher lung cancer rates, late diagnoses, and worse survival, with this discrepancy being most evident in Black patients [8]. The etiology of these disparities is multifactorial and often includes downstream effects of systemic racism such as education access, decreased access to healthcare, higher smoking prevalence, and limited economic opportunities [8,9]. In a comprehensive review of how lung cancer risk and structural racism are linked, Bonner et al. summarize seven main domains in the literature. They highlight that poor housing conditions disproportionately affect Black and Hispanic patients. Additionally, Black patients experienced higher occupational exposures to factors known to increase lung cancer risk. The impact of structural racism in policies contributes to increased poverty, decreased access to high-quality health insurance, as well as implicit medical biases [9]. Health literacy, often lower among individuals with low SES, further exacerbates challenges in accessing and understanding healthcare, affecting clinical decision-making and outcomes [9,10]. Ultimately, these social determinants perpetuate the disparities observed in lung cancer incidence and survival rates among disadvantaged populations. Addressing disparities in lung cancer screening, treatment, and survival could significantly reduce the morbidity and mortality burden of this disease.

## 2. Methods

This is a narrative review based on targeted literature searches conducted in PubMed and Google Scholar. We searched for studies published between 2000 and 2024 using combinations of terms such as lung cancer disparities, screening, treatment, genomics, and survival. Studies were selected for inclusion based on relevance, recency, and impact within each thematic section.

## 3. Lung Cancer Screening and Surveillance

Screening with an annual low-dose computed tomography (CT) scan can reduce lung cancer mortality by 16% to 24% among high-risk individuals by identifying asymptomatic cancers suitable for curative treatment [11]. However, overall screening rates remain low, with only about 6% of the 14.2 million eligible Americans screened annually [7]. Similar low screening rates are observed internationally [12]. Early detection through screening is crucial for improving lung cancer survival rates, yet significant disparities in screening eligibility and access persist [1,11,12,13].

New guidelines from the American Cancer Society recommend annual lung cancer screening (LCS) for healthy individuals aged 50 to 80 years with a smoking history of 20 pack-years or more, regardless of when they quit. This has expanded eligibility to an additional 5 million people and increased the potential to prevent lung cancer deaths [13]. However, these guidelines fail to address racial and ethnic differences. Data indicate that Black individuals develop lung cancer with fewer pack-years of smoking [13,14,15,16]. Black individuals are less likely to qualify for lung cancer screening due to lower tobacco exposure and younger age at diagnosis. Non-Black patients were 90% more likely to meet USPSTF (United States Preventive Services Task Force) criteria for lung cancer screening compared to Black patients. However, this disparity may be reduced with the use of more detailed risk-based screening criteria (PLCO_m2012_) [4,17,18]. Further existing data demonstrate that Black patients who are eligible for screening based on USPSTF criteria are less likely to be screened compared to White patients. Poulson et al. examined redlined areas in the Boston area. They found that Black patients had 61% lower odds of completing lung cancer screening than White patients in those areas despite meeting the USPSTF eligibility criteria at that time [19]. In another screening program in Philadelphia, Black patients had significantly lower odds of undergoing lung cancer screening compared to White patients after a referral. Among patients with a negative baseline screening CT, Black patients experienced longer intervals before subsequent screening compared to White patients and had higher rates of loss to follow-up [20].

The National Lung Screening Trial (NLST) found that White patients had statistically higher rates of follow-up care after screening tests compared to Black patients (89.6% vs. 82.8%; *p*  <  0.05) [10]. A study on LCS barriers among veterans found racial disparities, with Black veterans having lower odds of completing initial LCS compared to White veterans (30.5% vs. 41.3%). Key barriers included a higher proportion of current smokers among Black veterans (71.6% vs. 61.6%), a lower pack-year smoking history, and a higher prevalence of mental health or substance use diagnoses (65.6% vs. 54.6%). Despite living closer to VA screening centers, Black veterans faced lower screening completion rates [21]. In a larger study of the VA cohort, Black veterans had higher odds of delayed or no follow-up compared to White veterans [22].

Inequity in lung cancer screening is prevalent in other races and ethnicities as well. A study examining racial and ethnic differences in LCS completion and follow-up rates in Hawaii revealed significant disparities. Asian individuals had the highest completion rate at 86%, with Korean (94%) and Japanese (88%) subgroups having the highest screening rates. In contrast, lower rates were observed among Native Hawaiians (80%), non-Hispanic Whites (80%), Pacific Islanders (79%), and other racial and ethnic groups (77%), highlighting a 14% to 15% gap in LCS completion rates among the groups studied [23].

Disparities in LCS rates are often attributed to various factors [18,19,20,21,22,23,24,25,26,27,28,29,30] including insurance coverage. The Affordable Care Act mandates that most insurance plans cover LCS; however, Medicaid coverage varies. While 46 states’ Medicaid programs cover LCS, three do not. The Centers for Medicare and Medicaid Services provide LCS coverage for high-risk individuals aged 55–77. While Medicaid expansion has improved 2-year survival rates for non-older adult men with non-small-cell lung cancer (NSCLC) in expansion states, it has not had the same effect for women [24]. Income also plays a role, as studies reported that higher annual income is associated with higher completion rates or intent to undergo screening [25].

Geographic disparities also exist, with southern and western states having lower LCS completion rates [26]. One study highlighted that geographic disparities, including limited access to low-dose computed tomography facilities in rural and underserved areas, were more prevalent in these regions [26]. Another study conducted in rural settings found that 47% of participants cited financial constraints as a barrier to LCS. Thirty-five percent mentioned limited access to healthcare facilities, while 28% highlighted transportation issues. Additionally, 22% reported stigma and fear of false positives as deterrents [27]. Factors such as environmental or occupational exposures beyond tobacco use, access to care, transportation challenges, stigma and beliefs, difficulties in patient-practitioner communication, and shared decision-making all contribute to disparities in LCS [21,22,23,24,25,26,27,28,29,30].

Studies have reported differences in beliefs and attitudes among smokers. Black smokers were more likely than White smokers to believe they were at lower risk (28% vs. 15%) and that early detection would not improve survival (33% vs. 19%) [28]. Hispanic smokers expressed greater concern about screening costs (64% vs. 48% for Whites) and felt screening could cause unnecessary worry (47% vs. 34%) [29]. A qualitative study found that smoking-related stigma, characterized by shame and self-blame, was a major barrier to LCS. In a small group of majority White smokers, one cohort had been screened and the other had never been screened. They identified three major barriers to screening: inconvenience, smoking-related stigma, and distrust of the medical system [29].

## 4. Lung Cancer Preoperative Workup and Diagnosis

Accurate staging is pivotal in managing NSCLC, as it dictates both prognosis and treatment decisions [31,32,33,34]. Staging methods for NSCLC encompass both non-invasive and invasive techniques. Non-invasive methods, such as imaging modalities like CT and positron emission tomography (PET) scans, are widely employed for diagnosis and staging, but their utilization can vary considerably [32,33]. Guidelines from the American College of Chest Physicians and the National Comprehensive Cancer Network (NCCN) emphasize the role of PET scans in staging NSCLC patients eligible for curative treatments [31,32,33].

In a multi-site, prospective, observational study analyzing data from the Cancer Care Outcomes Research and Surveillance (CanCORS) Consortium between 2003 and 2005, PET scans were used 13% less often among non-White and Hispanic patients [32]. This disparity remained consistent across subgroups and could not be explained by variations in income, education, or insurance status. Although these data were published more than a decade ago, these findings persist. A more recent study using the national Surveillance, Epidemiology, and End Results (SEER)–Medicare linked database found similar results [34]. After accounting for demographic, community, and facility characteristics, Black individuals were still significantly less likely to undergo PET or CT imaging at the time of diagnosis compared to non-Hispanic Whites. Similarly, Hispanic individuals had a lower likelihood of receiving PET imaging [34].

## 5. Lung Cancer Genetic Testing

Genetic testing, fundamental to precision medicine, plays an essential role in diagnosing NSCLC and pinpointing mutations that guide the selection of targeted treatments [4,35]. Next-generation sequencing has identified numerous genetic anomalies in lung cancer, highlighting a panel of twenty genes with the highest mutation frequencies [4,35]. Based on a large Medicare study, molecular testing rates for NSCLC have risen in recent years but still fall short of recommended levels [36]. Despite increased molecular testing tools, disparities are pervasive in the use of this implement. In an analysis of the SEER database, 28,511 patients were identified with any stage of NSCLC [37]. Of these patients, 40.4% of White patients received molecular genetic testing compared to only 27.9% of Black patients (*p* < 0.001). There was not a sufficient number of patients of other races and ethnicities to perform additional comparison [37].

Despite a 19.7% increase in epidermal growth factor receptor (EGFR) testing from 2011 to 2013, Lynch et al. identified socioeconomic and geographic disparities in testing. Patients living in the northeastern regions had the highest likelihood of undergoing testing, and Asian/Pacific Islanders were the most likely racial group to undergo testing [38]. In contrast, Medicaid recipients, Hispanics, and Blacks were less likely to receive testing. Similarly, Illei et al. studied the trends and patterns of anaplastic lymphoma kinase (ALK) testing and found that patients with Medicare and Medicaid were less likely to undergo ALK testing [39]. However, unlike Lynch et al., they found that those residing in non-western states were also less likely to receive this testing. Using the SEER database, Kehl et al. reported molecular testing rates of 32.8% for Asian/other descents, 26.2% for Whites, and 14.1% for Blacks, with median survival times of 8.2 months for those tested and 6.1 months for those not tested [40]. Using data from the Kentucky Cancer Registry, Larson et al. found that factors like male sex, enrollment in Medicaid or Medicare, older age, geographic region, and smoking were linked to a lack of EGFR testing, and those tested had higher overall survival rates [41]. Interestingly, Cheng et al. discovered that among patients with EGFR-mutated NSCLC, Black patients had shorter survival rates compared to non-Black patients, with 2-year survival rates of 33% vs. 61%, respectively [42]. Begnaud et al. found no significant difference in mutation testing between American Indians and non-American Indians in Minnesota, suggesting that other factors contribute to higher mortality rates in this group [43].

Although broad-based genomic sequencing (BGS) has improved access to the genomic profile of advanced NSCLC tumors, as recommended by NCCN guidelines, disparities in testing still persist [35,36,37,38,39,40,41,42,43,44,45,46]. Riaz et al. observed significant age-related and racial disparities in the use of BGS among NSCLC patients treated in community settings. Black patients were less likely to receive BGS compared to White patients, and patients aged 76–85 were less likely to receive BGS compared to those under 45 years of age [44]. Likewise, Wu et al., in their analysis of real-world biomarker testing trends and overall survival among US patients with advanced NSCLC, reported that PD-L1 testing before first-line treatment was less frequently performed in older patients, those treated in a community setting, and those with squamous NSCLC, recurrent disease, or no distant metastasis. Testing was also less common, albeit to a lesser degree, among Black patients and those with Medicare insurance [45]. A recent systematic review of studies spanning 2007 to 2022 highlighted persistent racial and ethnic disparities in genomic testing for lung cancer patients, with Black patients consistently having lower testing rates compared to Asian and White patients [46].

A number of studies have examined the impact of race on lung cancer genetics but have reported conflicting results. In a prospective study of multiple Baltimore hospitals with an overrepresentation of Black patients, authors identified distinctly different elements in the NSCLC transcriptome when comparing African Americans to European Americans [47]. MicroRNA expression also varied between the two groups [47]. Another study analyzed over 509 lung cancers for the frequencies of genomic changes between Black and White patients, identifying 504 genes but finding no significant differences in genetic alterations between the groups [48]. With further advancements in genetic testing, it is crucial to improve genetic testing to identify any transcriptomic nuances present in patients.

Minority representation has been consistently lower in clinical trials and studies, despite minorities express a similar willingness to participate in medical research [49]. In an analysis of lung cancer clinical trials, Kanuparthy et al. found that although racial and gender minority enrollment has increased, it still has not reached parity across different races, ethnicities, and gender [50]. Similarly, Kwak et al. analyzed NSCLC patients from the National Cancer Database and found that Black and Hispanic patients were significantly less likely to be enrolled in lung cancer trials [51]. This underrepresentation was further confirmed in a more recent analysis of clinical trials enrolled on ClinicalTrials.gov (accessed on 10 April 2025) [52].

## 6. Lung Cancer Treatment

Disparities in lung cancer treatment have been well documented [53,54,55,56,57,58,59,60,61,62,63], with significant concerns regarding adherence to guideline-concordant treatment for minoritized groups. Notably, Black patients with resectable NSCLC are less likely to receive surgery and more likely to be treated with radiation therapy (RT) and/or chemotherapy—treatments not standard for this stage—even after adjusting for confounders such as comorbidities and age [53,54,55,56,57,58].

A SEER study found that Black patients underwent lung resection less frequently than White patients (69% vs. 83%, a 14% difference, 99% CI 11–18%), despite both groups being recommended to undergo surgical therapy [53]. Additionally, a notable delay from diagnosis to treatment for minorities has been reported [55]. Black patients were found to experience a longer time from diagnosis to treatment compared to White patients, with an approximate delay of eight days following multivariable analysis [55].

Contributing factors include socioeconomic vulnerabilities, limited access to experienced surgeons, misconceptions about surgery, and inadequate insurance coverage or reliance on public insurance. Distrust of healthcare providers is also a significant issue in this population [53,54,55,56,57,58,59]. Disparities in patient–surgeon interactions reveal that Black patients are less frequently offered surgery and have higher refusal rates when it is recommended [53,54]. This refusal is often linked to beliefs about accelerated tumor spread, potential for cure without surgery, and general distrust of healthcare providers [53]. Annesi and colleagues found that structural racism through residential segregation contributes to worse lung cancer outcomes for Black patients, including advanced disease at diagnosis, lower rates of early-stage surgery, and higher cancer-specific mortality, particularly in highly segregated areas [56].

A statewide quality collaborative involving high-volume centers in Michigan identified disparities in surgical quality for resectable NSCLC. Black patients received fewer lymph node resections and sampling than White patients. Unadjusted analysis found that Black patients were more likely to undergo wedge resection (limited resection); however, this difference was not observed in adjusted analyses. Further examination of the covariates found that Black patients were more likely to have lower predicted DLCO (diffusing capacity of the lung for carbon monoxide), higher rates of hypertension and diabetes, and a greater likelihood of requiring dialysis [58].

These disparities align with broader trends observed in national databases. Analysis from the SEER and New York Statewide Planning and Research Cooperative Database found that limited resection, increasingly used among patients with comorbidities (from 68% in the 1990s to 83% by 2007–2012), may lead to fewer in-hospital complications and shorter lengths of stay compared to lobectomy [59]. However, lobectomy with lymph node dissection is still regarded as the standard of care for the majority of operative lung cancer cases. Taioli and Flores further demonstrated that Black patients were significantly less likely to undergo surgery, and when they did, they were more likely to undergo a limited resection rather than a lobectomy compared to White patients [60]. Furthermore, Black patients had significantly less lymph nodes resected compared to White patients and had overall worse survival [60]. Similarly, SEER data show significant variability in resection rates for Black patients compared to White patients [61].

Specialty care is vital in thoracic oncology, with thoracic surgeons playing a pivotal role in evaluating and managing lung cancer patients. Cardiothoracic surgeons specializing in thoracic surgery perform the majority of minimally invasive (MIS) lobectomies compared to general surgeons or cardiac-track surgeons [61,62]. Notably, an analysis of the Premier database found that Black patients were less likely to undergo minimally invasive operations because they were less likely to be treated by thoracic surgeons [62]. A cohort analysis of the STS GTSD (Society of Thoracic Surgeons General Thoracic Surgery Database) also found that Black and Hispanic patients were less likely to undergo MIS surgery [63]. Additionally, Black patients, those with Medicaid insurance, and patients in smaller hospitals, the western region, and rural areas are less likely to have a thoracic surgeon perform their lobectomy, resulting in lower rates of MIS lobectomies for these groups [61,62].

## 7. Lung Cancer Survival

Lung cancer survival rates exhibit notable disparities across racial and ethnic groups despite advances in cancer awareness, screening, and treatment [4,64]. As shown by Wolf et al., Black patients with Stage I NSCLC were less likely to be treated with surgery and more likely to receive radiation or chemotherapy compared to their White counterparts. However, when analyzed according to treatment type, there were no differences in rate of survival [54]. Therefore, when provided comparable treatment, Black patients with Stage I NSCLC had similar rates of survival to White patients [54,56]. As highlighted by Wolf et al., socioeconomic factors, such as a lack of access to experienced surgeons, inadequate insurance coverage, and misconceptions associated with surgery, contribute to disparities in treatment [4,54,56]. Further, as discussed by Annesi et al., structural factors, such as residential segregation, exacerbate these disparities [56]. Studies have shown that Black patients residing in highly segregated areas are more likely to be diagnosed at an advanced stage and less likely to undergo surgical resection, leading to higher cancer-specific mortality rates [56]. This pattern underscores the role of structural racism in healthcare outcomes, where the geographic, socioeconomic, and cultural context significantly affects access to timely and appropriate care [4,54,56,64].

These disparities extend beyond Black patients and are present in other minority groups as well. Hispanic and Asian patients have been shown to have similar disparities, especially regarding the frequency of surgical interventions offered and the stage at diagnosis. Using the STS database, Weksler et al. found that for NSCLC, Hispanic and Asian patients had fewer operations for early-stage NSCLC than White and Black patients [65]. Hispanic and Asian patients were shown to have fewer surgical interventions and were diagnosed at a later stage compared to White patients [4,64,65]. These differences were reflected in the disparities in survival. Structural challenges, such as the impacts of immigration status, language concordance between providers and patients, and lack of cultural sensitivity, compounded with other structural barriers noted above, contribute to inferior survival outcomes [4,64,65].

The attribution of genetics as a key component in the differences in health outcomes is controversial and finds historical roots in eugenics [4,46,64]. While there may be certain genetic differences in lung cancers among races and ethnicities [66], these are not the sole reason why minorities are more likely to be diagnosed at later stages and have worse outcomes [67]. Further understanding of genetic differences is important for precision medicine. Additionally, studies analyzing genetic factors have repeatedly found that social determinants hinder access to diagnosis and treatment, often leading to later-stage presentation and substandard care, both of which have a more profound impact on outcomes and long-term survival [54,55,64]. Therefore, targeted interventions that consider the concepts of SDOH can work towards reducing racial disparities in lung cancer survival.

Survival disparities in patients with lung cancer are primarily driven by unequal access to timely diagnosis and the acceptance of appropriate guideline-directed treatment. These disparities are both compounded and influenced by socioeconomic, structural, and cultural barriers within the healthcare system and built environment that disproportionately affect minority populations. To improve survival outcomes and achieve equity in lung cancer care, it is essential to address these systemic issues, ensuring that all patients receive the best possible care regardless of their racial or ethnic background. This involves not only policy reforms and increased access to healthcare but also targeted efforts to overcome cultural and language barriers to provide culturally competent care.

## 8. Challenges and Future Opportunities

Race is a social construct. Thus, race-adjusted equations may increase health disparities by normalizing the differences in pulmonary function, limiting early diagnosis and intervention with pulmonary disease. In March 2023, the American Thoracic Society made an official recommendation to replace race- and ethnicity-specific equations with race-neutral reference equations for pulmonary function test (PFT) interpretation [68]. With these changes in how PFTs are calculated, Black patients’ FEV1 and DLCO measurements will appear lower (with a higher reference range in use). As a result, many patients previously calculated as having borderline function may end up being excluded from surgical intervention due to lower PFT calculations. It is unclear if these patients with borderline pulmonary function who previously underwent surgical intervention due to a perceived better function may have had better clinical outcomes with non-surgical intervention for their cancer. Providers need to be intentional about how these borderline patients are evaluated for surgical intervention and if PFTs are the best assessment for their surgical candidacy. There is an opportunity for continued research and education on how changes in PFT calculations alter surgical risk assessment.

The cultural factors that impact cancer outcomes can be challenging to address. For example, Black Americans have higher rates of medical distrust, potentially stemming from a history of medical experimentation and exploitation of Black Americans [29,53,54,55,56,57,58,59]. Among minority patients, there are higher rates of negative beliefs around standardized treatments compared to their White counterparts, even when patients are evaluated at high-volume centers and recommended for appropriate guideline-directed treatments [29,53,54,55,56,57,58,59]. Further investigations incorporating community-based research addressing barriers experienced by underserved groups need to be explored. Barriers such as low health literacy, medical mistrust, cancer fatalism, stigma, language barriers, time and transportation limitations, and cost constraints with insurance issues need to be studied and addressed [69]. It would be useful to evaluate local interventions in other diseases where providers have been successful in engaging the community. One suggested approach to bridging and mitigating some of these barriers is the utilization of Community Health Workers (CHWs) [70], by supporting the role of community partners who have pre-existing relationships with their communities [71].

A review of community engagement initiatives utilizing CHWs to promote cancer prevention and education in underserved communities in both high- and low–middle-income countries found that CHWs served as vital links between healthcare systems and the community they support. By providing culturally tailored education, outreach, and support, CHWs play crucial roles in breast cancer screening, including disseminating information, assisting with the screening process, and navigating patients through healthcare services. These initiatives led to significantly improved cancer screening rates and awareness which enhanced early detection and reduced barriers to care. Post-intervention evaluations demonstrated increased cancer awareness and screening rates, underscoring the effectiveness of employing CHWs in reducing cancer disparities [70]. A notable initiative in California, the “Fe en Acción” community intervention, significantly enhanced breast cancer screening rates among churchgoing Latinas [72]. The program, implemented in 8 of 16 randomly assigned churches, notably increased self-reported mammography (OR = 4.64) and breast exams (OR = 2.82), while also reducing barriers to breast cancer screening [72]. Efforts to implement CHWs in lung cancer screening have successfully increased awareness but face barriers to improving actual screening uptake. A pilot study in Alabama’s Black Belt highlighted challenges such as limited access to primary care providers, reluctance among providers to recommend screening, and logistical issues such as transportation and facility availability [73]. While CHWs effectively address knowledge gaps, systemic barriers must be overcome to translate awareness into action [73].

The U.S. Department of Health and Human Services created a priority area, “Healthy People 2030”, to mitigate the health disparities that exist by focusing on the domains of SODH [74]. As described earlier in this chapter, structural racism and residential segregation contributes to these disparities [56]. Distinguishing the individual roles of socioeconomic status, geography, and race as root causes of these disparities is difficult. The all-cause mortality of Black Americans with lung cancer is higher than that of their matched White counterparts. While some of this disparity is related to differences in cancer treatment, most excess mortality can be attributed to various SDOH beyond the scope of lung cancer diagnosis and treatment. These factors include occupational and environmental exposures, socioeconomic status, neighborhood characteristics, and health behaviors beyond smoking status [56].

One opportunity, albeit extraordinarily difficult to achieve, is improving access to high-quality medical care across racial and gender groups. Appropriate management and treatment of smoking-related chronic conditions and disease can help reduce the all-cause mortality of Black patients with lung cancer. One place to start is with individual groups, studying why there is unequal care presented to minority patients within a specific health system or community. For example, assessing the reason for loss of follow-up care for positive LCS imaging in minority patients within a specific healthcare system can help develop targeted interventions. Institutional assessments of inherent biases or factors that lead to lower guideline-concordant care or curative intent surgery in minority patients are critical. A major cause of disparity in Black and Hispanic patients is limited access to experienced surgeons and high-volume centers—disparities that lead to treatment delay and higher morality [62]. These disparities highlight underlying societal issues that need to be addressed and the importance of moving away from a one-size-fits-all approach. Instead, health policies should focus on investing in tailored strategies that assess and address the unique risk factors and needs of individual communities, acknowledging their distinct challenges and priorities.

## 9. Intervention Strategies

There are several proposed solutions for improving health disparities in the literature, many of which emphasize multilevel approaches that target change across the healthcare ecosystem (Figure 2). Black patients are diagnosed with lung cancer at younger ages with fewer pack-years of smoking than their white counterparts. As a result, the current USPSTF guidelines (ages 50–80 with 20 pack-year history) may fail to capture all high-risk individuals. Creating LCS eligibility criteria that accurately reflect the cancer risk across the different racial and ethnic groups is the key to reducing screening disparities [18]. A change in formal LCS recommendations to include a risk assessment algorithm that includes race, smoking duration and intensity, environmental factors, and socioeconomic status may better capture higher-risk individuals [75]. Potter et al. proposed changes to the USPSTF guidelines. In their analysis of the Southern Community Cohort, the authors found that significantly more White patients would have undergone lung cancer screening based on the current USPSTF guidelines (57.6% of Black patients vs. 74% of White patients, *p* < 0.001) [76]. The authors proposed using a 20-year smoking duration rather than a 20-pack-year smoking cutoff. With this change, the percentage of Black and White patients who qualified for screening were similar (85.3% vs. 82%). The authors found a similar trend when using their proposed guidelines in the Black Women’s Health Study. Of the patients with lung cancer, 63.8% would have qualified for screening as opposed to the 42.5% (*p* < 0.001) with the USPSTF guidelines [76].

A system-based intervention can also serve as a potential solution. An example of this is the multi-institutional “ACCURE” study, which implemented a system-based intervention to address disparities in cancer treatment completion and outcomes. This intervention included a real-time warning system based on electronic health records, feedback to clinical teams on treatment disparities by race, and the support of a nurse navigator [76]. By comparing patients from 2013–2015 to historical controls from 2007–2011, the study found significant improvements in survival rates for both Black and White patients with breast cancer and White patients with lung cancer. The intervention reduced the racial gap in survival rates and improved treatment completion across the board. Despite these gains, some disparities remained, particularly in the selection of treatment modalities for lung cancer, suggesting a need for further investigation [76].

Cultural barriers can be difficult to overcome at a provider level, but medical-adjacent support staff may help in this arena. One place to start is in the clinic, with clear communication between the patients and providers via the use of certified interpreters as well as written educational material available in the patient’s primary language that is culturally appropriate [77,78,79]. Another possibility to improve care in this realm is leveraging the lay individuals in the community and trusted medical providers to help bridge knowledge gaps, alleviate patient concerns, and promote increased screening engagement and trust in the healthcare system [70,72]. Additionally, provider training on culturally competent care is critical to staff education [80].

Factors associated with low socioeconomic status that contribute to worse outcomes can be mitigated through several deliberate interventions. Local interventions to improve care compliance of patients with limited resources include transportation vouchers, free or discounted parking, appointments on weekends and evenings, care coordination (of labs, imaging, and provider visits) to minimize hospital trips, and appointments closer to home or virtual visits [77,81,82,83,84,85,86,87,88,89,90]. Other efforts that can mitigate access barriers that have shown some success are mobile screening clinics [91,92]. These programs deploy mobile low-dose CT units to community locations, offer on-site screenings to reduce travel burdens, and conduct targeted outreach to educate high-risk groups [91]. Recently proposed legislation supported by the Society of Thoracic Surgeons aims to expand the use of mobile lung cancer screening units. The bill establishes a competitive grant program within the health resources and services administration, emphasizing their role in reducing disparities and improving early detection for high-risk and underserved communities [92]. The use of patient navigators and social workers has been shown to help patients with limited health literacy complete medical recommendations [89]. State-funded screening programs can also help alleviate the cost associated with LCS [90].

Structural racism remains a difficult long-term problem. Implicit bias training and cultural communication training have been successful in improving care in other realms of healthcare [93,94]. Designing standardized staging and treatment protocols both nationally and locally at high-volume centers may minimize the effect of implicit bias. National- and local-level changes requiring equal representation in research at all levels of lung cancer will improve enrollment and participation of underrepresented groups in clinical trials and studies.

## 10. Conclusions

Addressing disparities in lung cancer care is essential to improving outcomes and achieving health equity for all patients. These disparities are driven by complex factors, including social determinants of health, systemic racism, and limited access to high-quality care. Targeted interventions, such as risk-based screening guidelines, culturally competent care, community-based initiatives, and policy reforms, are crucial to reducing these inequities. Prioritizing inclusivity, enhancing access to care, and implementing tailored strategies, such as real-time tracking tools, patient navigation, and mobile screening units, can help bridge gaps in care.

## Figures and Tables

**Figure 1 cancers-17-01347-f001:**
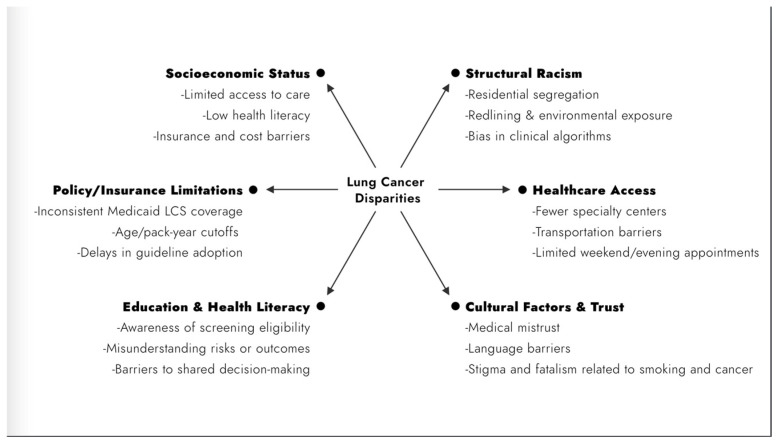
Social determinants of health contributing to lung cancer disparities. LCS, lung cancer screening.

**Figure 2 cancers-17-01347-f002:**
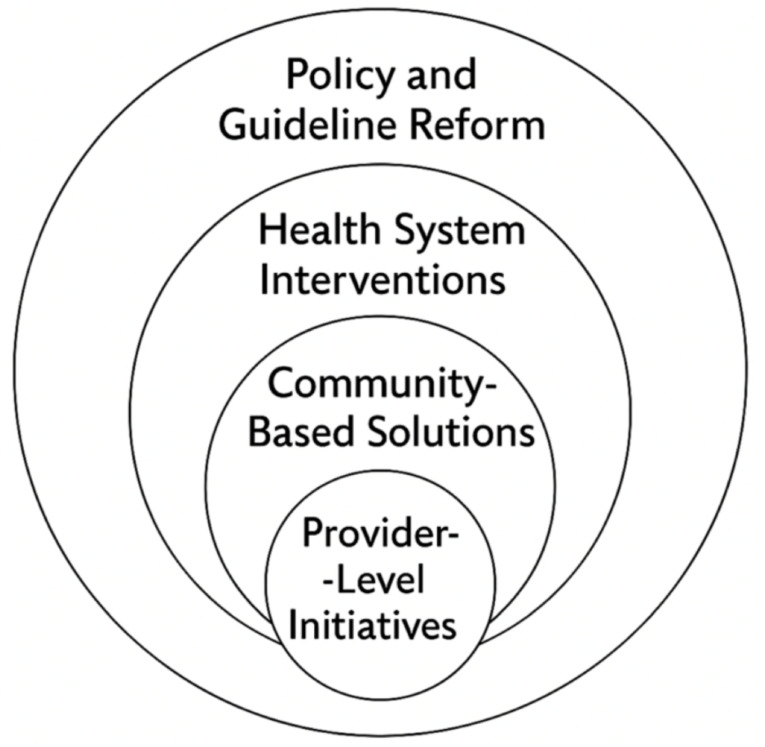
Intervention targets to address lung cancer care disparities.

## Data Availability

No new data were created or analyzed in this study. Data sharing is not applicable to this article.

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
