# Peer review of "Equity and Opportunities in Lung Cancer Care—Addressing Disparities, Challenges, and Pathways Forward"

_cancers, 2025, doi:10.3390/cancers17081347_

Round 1

Reviewer 1 Report

Comments and Suggestions for Authors

The article is well-designed and well-structured.

I am missing a small methodology section on how they have selected these papers to be able to do a quality review. How many did you find? How many did you discard based on the selection criteria, for example?

The English is correct, and the references are up to date. However, I repeat we do not know if any important ones have been discarded because we do not know the criteria they have selected.

Therefore, it would be interesting to add a small explanatory section.
In Spain we do not have such a black population, but if there is evidence that says “depending on your zip code”, you will have one type of health care or another. Unfortunately, health policies are needed to detect these deficiencies to change the health status of certain population groups, as they rightly comment.

I encourage authors to add a touch of quality by adding how they made the selection in order to create this review.

Best regards

Author Response

Reviewer: I am missing a small methodology section on how they have selected these papers to be able to do a quality review. How many did you find? How many did you discard based on the selection criteria, for example?

Response: Thank you for your thoughtful feedback. We have added a brief methodology section to clarify our approach. As this was a narrative review, we used a broad search strategy and snowballing to identify relevant literature. While not systematic, our goal was to capture key themes and widely cited work in this area. We appreciate your insights on geographic disparities and policy implications.

Reviewer 2 Report

Comments and Suggestions for Authors

Authors in their review entitled “  Equity and Opportunities in Lung Cancer Care —Addressing Disparities, Challenges, and Pathways Forward “ summarize the state of art on Lung Cancer Inequalities in United States.

The article is well written and authors are well known in the field.

I recommend for the publication after minor revisions.

I suggest to include two figures that highlight some key concepts:

  1. a) Social determinants that contribute to lung cancer health disparities;

-b) strategies to improve lung cancer care in disadvantages ethnical groups.

Insert some references:

-JAMA Netw Open. 2024;7(5):e2412880. doi:10.1001

-Public Healthhttps://doi.org/10.1016/j.puhe.2024.08.021

Author Response

Comment 1: 

I suggest to include two figures that highlight some key concepts:

  1. a) Social determinants that contribute to lung cancer health disparities;

-b) strategies to improve lung cancer care in disadvantages ethnical groups.

Response: Thank you for the insightful suggestion. We have added a figure illustrating the social determinants contributing to lung cancer disparities (figure 1). In response to point (b), we included a second figure highlighting multilevel targets for interventions to improve care among disadvantaged populations. We appreciate your feedback.

Comment 2: Insert some references:

-JAMA Netw Open. 2024;7(5):e2412880. doi:10.1001

-Public Healthhttps://doi.org/10.1016/j.puhe.2024.08.021

Response: Thank you for the suggestion to strengthen the manuscript with additional references. We have incorporated the recommended citations (5 and 69). 

Round 2

Reviewer 1 Report

Comments and Suggestions for Authors

In my case, they have answered the questions and added the missing information. I would accept the job.
Thank you